# Temporal-Oriented Recipe for Transferring Large Vision-Language Model to Video Understanding

## Abstract

Recent years have witnessed outstanding advances of large vision-language models (LVLMs). In order to tackle video understanding, most of them depend upon their implicit temporal understanding capacity. As such, they have not deciphered important components that contribute to temporal understanding ability, which might limit the potential of these LVLMs for video understanding. In this work, we conduct a thorough empirical study to demystify crucial components that influence temporal understanding capacity of LVLMs. Our empirical study reveals that significant impacts are centered around the intermediate interface between the visual encoder and the large language model. Building on these insights, we propose a temporal-oriented recipe that encompasses temporal-oriented training schemes and an upscaled interface. Our final model developed using our recipe significantly enhances previous LVLMs on standard video understanding tasks. [1]

## 1 Introduction

Empowered by the elevating popularity of video-text data (Nguyen et al., 2024c;b) and outstanding advances in large language model (LLM)-based designs, recent years have encountered remarkable progress in video understanding with large vision-language models (LVLMs). From the advent of models such as BLIP (Li et al., 2022), BLIP-2 (Li et al., 2023a), and LLaVA (Liu et al., 2023), video question answering (VideoQA) has improved from 33.8, 16.7, and 12.4 on MSVD (Wu et al., 2017), MSRVTT (Xu et al., 2016), and ActivityNet (Krishna et al., 2017) to more than 60.0 in terms of GPT-3.5 evaluation. Not only VideoQA but also long-term action recognition (Kuehne et al., 2014; Tang et al., 2019; Wu & Krahenbuhl, 2021) and video captioning (Zhou et al., 2018; Islam et al., 2024) have achieved significant breakthroughs.

In recent years, model architectures and training protocols have witnessed significant advancements. As these systems grow in diversity and scale, their computational demands pose substantial challenges for comparison, analysis, and reproducibility. Despite these advancements, many approaches have overlooked the core nature of video understanding. Rather than explicitly modeling temporal relationships, they often rely on spatial inductive biases, assuming that spatial knowledge can seamlessly extend to temporal comprehension. For instance, several methods focus on creating a unified representation space for visual and textual modalities (Lin et al., 2023; Chen et al., 2023; Zhang et al., 2023). Others emphasize aggregating or selecting salient visual tokens aligned with prompts (Shang et al., 2024; Xu et al., 2024) or leverage large-scale pretraining with instruction-following datasets (Maaz et al., 2023; Luo et al., 2023; Wang et al., 2024b). Therefore, existing models fall short of realizing the full potential of video understanding. For example, while VideoQA systems, *e.g.* Video-LLaMA (Zhang et al., 2023), Video-LLaVA (Lin et al., 2023), and Qwen2.5-VL (Bai et al., 2025), can accurately answer questions about object detection or describe isolated actions, they struggle with queries involving causal and temporal relationships (Xiao et al., 2021). As shown in Table 1, they often generate inaccurate responses when faced with questions about temporal order or causality.

To overcome this limitation, we aim to enhance temporal understanding capabilities of large vision-language models (LVLMs) by advancing temporal-critical components within their architectures.

---

[1] Our codes and data are available at https://github.com/... (the link is hidden now due to double-blind review)

Table 1: On the first row, a correct answer should comprise details related to cutting ginger and garlic on a chopping board, whereas other models wrongly mention "*rub salt*", "*cut chicken*" and "*add to the pot*", and "*pour milk*". On the second row, we need to respond with "*getting to the bus*", but the models mistakenly note "*late for exam*", "*to the hospital*", and "*feeling sad*".

| Video | Question | Sample Answer | Video-LLaMA | Video-LLaVA | Qwen2.5-VL |
|---|---|---|---|---|---|
| | What does the person do before seasoning chicken? | The person cuts ginger and garlic into small pieces on a chopping board | The person rubs salt on the chicken. | Before seasoning chicken, the person cuts it into small pieces and adds it to a pot. | Before seasoning chicken, the person pours milk into a bowl of rice. |
| | Why did the girl rush? | The girl rushed to get to the bus on time, as she was running to catch the bus | The girl rushed because she was late for her exam. | The girl rushed to the hospital because she was feeling unwell. | The girl in the picture appears to be looking down and might be feeling sad or contemplative. |

Table 2: Existing LVLM models exhibit stark distinctions among themselves, making it challenging to reproduce, analyze, and compare these methods. Therefore, we aim to answer the question: "*Is there a straightforward recipe to build temporal understanding capacity for LVLMs?*"

| Method | Visual Encoder | Vision-Language Interface | LLM Size | Training Data Size (Pretraining) | Training Data Size (IFT) |
|---|---|---|---|---|---|
| VALLEY (Luo et al., 2023) | ViT-L | Transformer + Mean-pooling | 13B | 702K | 73K |
| Video-LLaMA (Zhang et al., 2023) | CLIP-G | Q-Former | 7B | 3M | 18K |
| LLaMA-VID (Li et al., 2024b) | CLIP-G | Linear Projection | 13B | 790K | 763K |
| VideoChat (Li et al., 2023b) | CLIP-G | Q-Former | 7B | 25M | 18K |
| VideoChat2 (Li et al., 2024a) | UMT-L | Q-Former | 7B | 25M | 2M |
| Video-ChatGPT (Maaz et al., 2023) | ViT-L | Mean-pooling + Linear Projection | 7B | 595K | 100K |
| Video-LLaVA (Lin et al., 2023) | ViT-L | Linear Projection | 7B | 1.3M | 765K |
| GPT4Video (Wang et al., 2024b) | ViT-L | Q-Former | 7B | 11K | 50K |
| PLLaVA (Xu et al., 2024) | ViT-L | Linear Projection + Adaptive Pooling | 7B/13B/34B | 25M | 783K |
| ST-LLM (Liu et al., 2024a) | BLIP-2 | Linear Projection | 7B | 25M | 2M |
| Chat-UniVi (Jin et al., 2024) | ViT-L | Clustering-based Merging + Linear Projection | 7B/13B | 1.6M | 649K |

As illustrated in Figure 1, an LVLM is fundamentally composed of three main components: a visual encoder, a vision-language interface, and a large language model (LLM). However, due to the large-scale nature of LLMs and the multimodal complexity of video data, identifying the primary factors driving model effectiveness is challenging (He et al., 2024; Qian et al., 2024; Chandrasegaran et al., 2024), hindering further progress in the field. Our focus is to bridge this gap by ensuring that temporal understanding is treated as a core aspect of video comprehension, rather than an implicit outcome of spatial knowledge.

To illustrate our points, in Table 2, we explicate the diversity of modern LVLMs for video understanding by examining them along various dimensions, including visual encoder, vision-language interface, LLM, and training data. Based on this examination, we observe that there exist stark differences among these models. Nevertheless, it is not straightforward to dissect which factors make an important contribution to overall video understanding performance and which do not.

As one of the works that initiates empirical analysis research line, METER (Dou et al., 2022) and MM1 (McKinzie et al., 2024) study a wide variety of modular components and pretraining design choices in the context of image-language modeling. Unfortunately, its analysis mostly works on images and neglects many aspects related to video modeling, such as spatio-temporal architectural design, video pretraining data, and video pretraining objectives. To fill in such gap, recent VindLU work (Cheng et al., 2023) conducts an analysis towards important factors for video-language understanding models. Unfortunately, their analysis is limited to small-scale frameworks with millions of parameters. Similarly, Fu et al. (2023) performs an empirical study of video-language transformers, but narrowly concentrates on masked visual modeling objective.

Our primary objective in this work is to answer the question "*Is there a straightforward recipe to build temporal understanding capacity for LVLMs?*" Our answer is **yes**. To arrive at the answer, we conduct a thorough empirical study that demystifies the importance of various design choices and ultimately leads to a temporal-oriented recipe that significantly enhances video understanding results of previous LVLMs. Our recipe starts from a standard paradigm of a large vision-language model then proceeds with a progressive expansion scheme, where at each stage, we investigate a specific aspect of LVLM framework design (*e.g.*, architecture, training objective, training data, etc.) and choose the most effective option. Particularly, we study the following LVLM design components: (i) the vision-language interface, (ii) the video training protocols, and (iii) temporal memory bank, and (iv) scaling of the essential component. We present our recipe in Figure 1.

Figure 1: Our temporal-oriented recipe for large vision-language model.

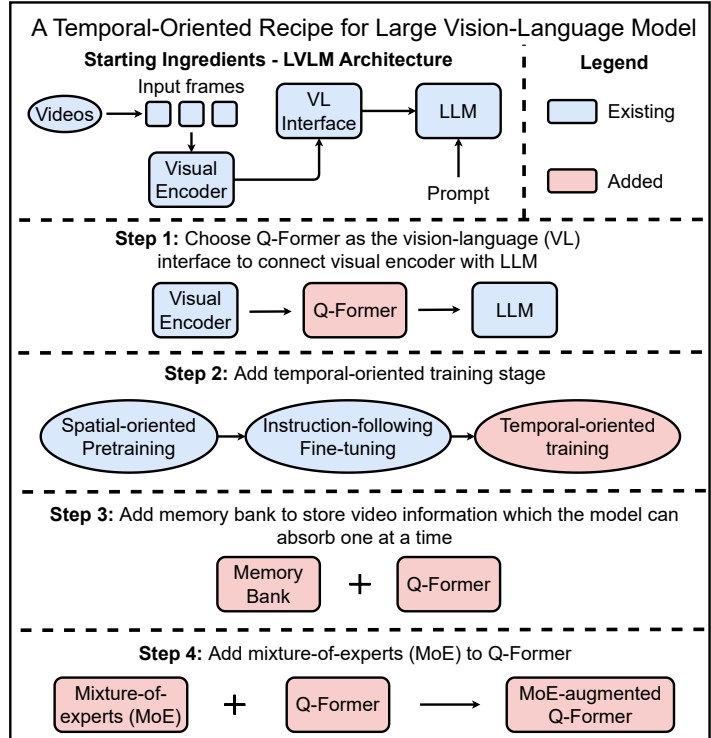

The key lessons of our study include:

- Among components in an LVLM architecture, we discover that enhancing vision-language interface significantly advances the temporal modeling strength of the LVLM.
- A query transformer that incorporates query tokens to interact with video representations combined with a temporal memory bank to compress salient video information is crucial for satisfactory video understanding performance.
- We can further obtain gains of temporal understanding level with techniques to scale up the interface, including mixture-of-experts and number of query tokens to store video information.
- An additional training stage for LVLMs with temporal-oriented data is sufficient to remarkably enhance temporal understanding capability and achieve impressive results.

## 2 RELATED WORK

### 2.1 LARGE VISION-LANGUAGE MODELS

Recent progress in LLM-based methods has produced large vision-language models (LVLMs) with strong visual understanding (Luo et al., 2023; Zhang et al., 2023; Li et al., 2024b; 2023b; 2024a; Maaz et al., 2023; Lin et al., 2023; Wang et al., 2024b; Xu et al., 2024; Liu et al., 2024a; Jin et al., 2024). Most follow a similar paradigm of visual encoder, vision-language interface, and LLM, though design differences make it difficult to identify key performance factors. Empirical studies such as VindLU (Cheng et al., 2023) and METER (Dou et al., 2022) provide insights, but focus on medium-scale or image-only settings. Our work instead examines billion-parameter models explicitly tailored to video understanding, offering clearer guidance for future LVLM design.

### 2.2 VIDEO UNDERSTANDING

Video understanding has advanced rapidly (Li et al., 2023b; 2024a; Nguyen et al., 2025a;b; Chen et al., 2023; Lin et al., 2023), achieving strong results on captioning and QA (Chen et al., 2023; Wang et al., 2024a; Maaz et al., 2023). LVLMs typically bridge the visual encoder and LLM via linear projection, Q-Former, or temporal encoder (Zhang et al., 2023; Li et al., 2024b; Lin et al., 2023; Xu et al., 2024;

Jiang et al., 2025). Eagle-2.5 (Chen et al., 2025) further improves long-context modeling with tiling and progressive training. Training commonly relies on mixed image- and video-text datasets, which complicates separating spatial vs. temporal reasoning. Unlike image models, there remains little systematic analysis of which components are essential for temporal reasoning in LVLMs.

## 3 TEMPORAL-ORIENTED RECIPE FOR LARGE VISION-LANGUAGE MODEL

In this section, we delineate our temporal-oriented recipe for large vision-language model. We start with a standard large vision-language model (LVLM), which consists of a visual encoder such as ViT and a large language model (LLM). Then, we progressively expand it to a model that achieves impressive temporal understanding results on various video understanding datasets and tasks. At each step of our recipe, we investigate how design choices have an impact upon temporal capacity of the LVLM. Throughout our procedure, we will discover answers to the following questions about the temporal-oriented recipe design:

- Can explicitly constructing temporal understanding capacity help LVLM, particularly provided that various video understanding benchmarks are spatially biased (Buch et al., 2022; Lei et al., 2022)? If so, what is the best mechanism for LVLM to conduct temporal modeling?

- Given that video lengths vary with a wide range, what is the most productive mechanism for LVLM to read/absorb video information? Several approaches use visual encoder combined with linear projection (Maaz et al., 2023; Xu et al., 2024) or query transformer (Q-Former) (Li et al., 2023b; 2024a), then proceed with a pooling mechanism, whereas others adopt a memory bank (He et al., 2024). Which of these is the most effective?

- Which temporal-oriented training schemes are most useful for temporal representation learning? There exist a wide variety of schemes, including video captioning (Abdar et al., 2024), moment captioning (Qasim et al., 2025; Yang et al., 2023), moment grounding (Nguyen et al., 2023b; Lei et al., 2021), and video summarization (Apostolidis et al., 2021; Nguyen et al., 2023a). How significant is each of these schemes? Are they complementary to each other?

- How can we optimize temporal capacity of LVLM? Can we inherit the mixture-of-experts (MoE) approach from LLM works?

STEP 0: STARTING INGREDIENTS

**Large Vision-Language Model.** We start with a standard ViT-G/14 (Dosovitskiy et al., 2020) from EVA-CLIP (Fang et al., 2023). For LLM, we use either Vicuna-7B or Vicuna-13B (Chiang et al., 2023), forming either a 7B-LVLM or a 13B-LVLM, respectively. Formally, given a paired video and text prompt $(V, T)$, the visual encoder randomly selects a sequence of frames from the video as input to extract visual embeddings. The LLM encodes the prompt $T$ to extract the textual embeddings.

**Experimental Setup.** As our initialization, we directly inherit the pretrained and instruction-tuned model on image-based data (Liu et al., 2023; Li et al., 2022; 2023a). Afterwards, we either conduct an additional temporal-oriented training step or go straight to finetuning and evaluating the model on the seven popular video understanding datasets: MSRVTT (Xu et al., 2016), MSVD (Chen & Dolan, 2011), ActivityNet-QA (Caba Heilbron et al., 2015), Breakfast (Kuehne et al., 2014), COIN (Tang et al., 2019), and LVU (Wu & Krahenbuhl, 2021). For our empirical investigation, we choose the video question answering (VideoQA) task and report the accuracy across these datasets.

In the following subsections, we progressively expand this baseline by adding components of elevating complexity. Specifically, we start by incorporating vision-language interface (step 1), then integrate a temporal-oriented training stage (step 2), insert a temporal memory bank (step 3), and upscaling the interface (step 4). Due to large computational cost, we cannot ablate the order of the steps in our recipe. Therefore, the order of the steps is primarily determined by the computational cost (*i.e.* the steps that can be implemented most efficiently are investigated before other steps, subsequently moving to more computationally costly steps).

Table 3: Effect of different types of vision-language interface on 7B-LVLM

| Vision-Language Interface | MSRVTT | MSVD | ActivityNet-QA | Breakfast | COIN | LVU |
|---|---|---|---|---|---|---|
| Linear Projection | 45.3 | 56.9 | 46.3 | 85.9 | 83.9 | 57.1 |
| Q-Former w/o SA + Mean-pooling - S = 3 | 45.8 | 57.4 | 46.4 | 86.3 | 84.8 | 58.0 |
| Q-Former w/o SA + Adaptive-pooling - S = 3 | 46.2 | 57.5 | 46.8 | 87.0 | 85.6 | 58.2 |
| Q-Former w/o SA + ESA - S = 3 | 46.3 | 57.7 | 47.2 | 87.6 | 86.0 | 58.7 |
| Q-Former w/o SA + Mean-pooling - S = 6 | 46.3 | 58.1 | 47.6 | 87.9 | 86.5 | 58.8 |
| Q-Former w/o SA + Adaptive-pooling - S = 6 | 46.6 | 58.1 | 47.6 | 88.3 | 87.1 | 59.2 |
| Q-Former w/o SA + ESA - S = 6 | 47.0 | 58.2 | 48.1 | 88.4 | 87.6 | 59.8 |
| Q-Former w/o SA + Mean-pooling - S = 9 | 47.2 | 58.2 | 48.3 | 88.5 | 87.9 | 60.3 |
| Q-Former w/o SA + Adaptive-pooling - S = 9 | 47.4 | 58.7 | 48.6 | 89.1 | 88.5 | 60.8 |
| Q-Former w/o SA + ESA - S = 9 | 47.5 | 59.1 | 48.8 | 89.9 | 88.6 | 61.2 |
| Q-Former w/o SA + Mean-pooling - S = 12 | 47.6 | 59.2 | 49.3 | 90.1 | 89.1 | 61.8 |
| Q-Former w/o SA + Adaptive-pooling - S = 12 | 47.9 | 59.8 | 49.5 | 90.8 | 89.4 | 62.2 |
| Q-Former w/o SA + ESA - S = 12 | 48.2 | 59.9 | 49.5 | 90.9 | 90.0 | 62.4 |
| Q-Former w/ SA - S = 3 | 48.7 | 60.2 | 49.8 | 91.7 | 90.8 | 62.7 |
| Q-Former w/ SA - S = 6 | 48.8 | 60.4 | 49.8 | 91.9 | 91.6 | 52.0 |
| Q-Former w/ SA - S = 9 | 49.0 | 60.4 | 50.3 | 92.1 | 92.5 | 63.5 |
| Q-Former w/ SA - S = 12 | 49.3 | 60.4 | 50.3 | 92.4 | 92.7 | 63.7 |
| Pre-trained Q-Former w/ SA - S = 12 | **49.4** | **60.8** | **50.6** | **93.1** | **93.4** | **63.8** |

Table 4: Effect of different types of vision-language interface on 13B-LVLM

| Vision-Language Interface | MSRVTT | MSVD | ActivityNet-QA | Breakfast | COIN | LVU |
|---|---|---|---|---|---|---|
| Linear Projection | 56.2 | 67.9 | 48.7 | 86.5 | 84.5 | 65.2 |
| Pre-trained Q-Former w/ SA - S = 12 | 56.8 | 68.2 | 48.7 | 86.6 | 85.1 | 67.2 |
| Q-Former w/o SA + Mean-pooling - S = 3 | 56.9 | 68.3 | 49.0 | 87.2 | 85.2 | 67.3 |
| Q-Former w/o SA + Adaptive-pooling - S = 3 | 57.1 | 68.7 | 49.3 | 87.7 | 86.0 | 67.6 |
| Q-Former w/o SA + ESA - S = 3 | 57.5 | 69.3 | 49.5 | 88.0 | 86.0 | 68.1 |
| Q-Former w/o SA + Mean-pooling - S = 6 | 57.8 | 69.7 | 49.5 | 88.4 | 86.4 | 68.1 |
| Q-Former w/o SA + Adaptive-pooling - S = 6 | 58.0 | 69.8 | 49.8 | 88.7 | 87.3 | 68.5 |
| Q-Former w/o SA + ESA - S = 6 | 58.1 | 70.4 | 50.0 | 88.8 | 87.8 | 69.0 |
| Q-Former w/o SA + Mean-pooling - S = 9 | 58.2 | 71.0 | 50.2 | 89.3 | 88.2 | 69.1 |
| Q-Former w/o SA + Adaptive-pooling - S = 9 | 58.3 | 71.2 | 50.3 | 89.4 | 88.9 | 69.4 |
| Q-Former w/o SA + ESA - S = 9 | 58.7 | 72.8 | 50.4 | 90.2 | 89.5 | 69.6 |
| Q-Former w/o SA + Mean-pooling - S = 12 | 59.1 | 72.2 | 50.5 | 90.6 | 90.2 | 69.8 |
| Q-Former w/o SA + Adaptive-pooling - S = 12 | 59.2 | 72.3 | 50.6 | 91.3 | 90.7 | 70.1 |
| Q-Former w/o SA + ESA - S = 12 | 59.5 | 72.5 | 51.1 | 92.1 | 90.9 | 70.3 |
| Q-Former w/ SA - S = 3 | 59.8 | 72.5 | 51.5 | 92.4 | 91.1 | 71.0 |
| Q-Former w/ SA - S = 6 | 59.9 | 73.1 | 51.7 | 92.4 | 91.5 | 71.2 |
| Q-Former w/ SA - S = 9 | 60.3 | 73.8 | 51.8 | 92.9 | 92.0 | 71.5 |
| Q-Former w/ SA - S = 12 | 60.5 | 74.3 | 52.0 | 93.7 | 92.4 | 71.5 |
| Pre-trained Q-Former w/o SA - S = 12 | **60.6** | **74.3** | **52.5** | **93.7** | **93.2** | **71.8** |

STEP 1: VISION-LANGUAGE INTERFACE FOR LVLM

In the first stage of our temporal-oriented recipe, we investigate the interface between the vision and language domain for our LVLM. Such interface will enable the LLM to have access to visual information from the video input. For compactness, we study three interface schemes:

- **Linear projection:** In this interface, the linear projection maps visual embeddings into appropriate dimensional space for the LLM. Due to its simplicity, this approach has been widely adopted by previous LLaVA-based LVLMs (Xu et al., 2024; Lin et al., 2023).

- **Query Transformer with Self-Attention (Q-Former w/ SA):** Following (Zhang et al., 2023; He et al., 2024), we use a number of transformer submodules which consist of cross-attention and self-attention layers. Cross-attention layers will enable a set of learnable query embeddings to interact with video representations to extract video information. In this variant, our Q-Former also contains self-attention layers, which can perform temporal modeling since they relate video frames together. We vary the number of submodules $S \in \{3, 6, 9, 12\}$. Parameters of Q-Former can be either randomly initialized or initialized from a pre-trained model. In our work, if we initialize Q-Former from a pre-trained model, we follow MA-LMM (He et al., 2024) to use the *bert-base-uncased* with $S = 12$.

- **Query Transformer without Self-Attention (Q-Former w/o SA):** This version is similar to the previous one, except the fact that Q-Former does not comprise self-attention layers. Therefore, we need to incorporate an additional component after Q-Former for temporal

modeling. We experiment with possible choices, including mean-pooling, adaptive pooling, and external self-attention (ESA) layers.

As Table 3 and 4 show, Q-Former demonstrates critical performance improvement over the linear projection approach. The improvement is indicated by average +6.0% and +6.2% accuracy boost of our 12-layer Q-Former variant over the 7B and 13B linear-projection baseline, respectively. We also observe that initializing Q-Former self-attention layers with pretrained BERT encoder makes a significant contribution to the performance boost. This suggests that temporal semantics among words can be related to temporal relations among video frames.

Interestingly, our findings contradict the conclusions of several prior studies (Liu et al., 2023; Koh et al., 2023), which suggest that a simple linear projection is sufficient, and even more effective, than the Q-Former approach. In contrast, we observe that Q-Former plays a crucial role due to its ability to model diverse temporal relations across a broad range of video scenarios. We hypothesize that, particularly for temporally-intensive datasets, the integration of stacked cross- and self-attention layers provides the necessary capacity to capture and reason about complex temporal dependencies across video frames.

*Takeaway 1: For all subsequent experiments, we use 12-layer pretrained Q-Former w/ SA as our vision-language interface for video understanding with large vision-language model (LVLM).*

STEP 2: TEMPORAL-ORIENTED TRAINING SCHEMES

Existing methods (Zhang et al., 2023; Li et al., 2024b; Lin et al., 2023) typically follow a pipeline of pretraining and instruction-tuning, followed by downstream finetuning. In our work, we investigate whether introducing an additional training stage specifically aimed at enhancing temporal understanding can further improve the video comprehension capabilities of LVLMs. To this end, we explore several temporal-oriented training strategies, which are illustrated in Table 7.

- **Video Captioning (VC):** aims to generate compact content of the video by leveraging the encoded information from the video. This objective resembles the next token prediction scheme to pretrain text-only LLM. To implement this objective, we provide the LVLM with a video input and the prompt "*what does the video describe?*", then train it to generate the groundtruth caption. For training data, we utilize 661K video-text pairs from 10M samples of the VIDAL-10M dataset (Zhu et al., 2023).

- **Moment Captioning (MC):** Slightly different from VC, MC aims to caption only a specified part of the video. To implement this objective, we leverage the 745K samples from the InternVid dataset (Wang et al., 2023), each of which consists of a query and the specific starting and ending timestamps of the related moment in the video. Based on these timestamps, we convert them to discrete frame indices, then provide the model with the prompt "*Explain what happened from frame <start> to frame <end> in the video.*"

- **Moment Grounding (MG):** The MG task is the reverse variant of MC. Instead of training the model to write a caption, we let it generate the indices of the start and end frame index of the moment caption. Analogous to MC, we also employ the 745K samples from the InternVid dataset (Wang et al., 2023).

- **Dense Captioning (DC):** This task is the more complete and fine-grained version of MC and VC, respectively. In particular, we ask the LVLM "*Can you give me a breakdown of the occurrences at different timestamps in the video?*". As Table 7 shows, the model is expected to describe a list of moments with the respective frame indices related to the moment.

Based on the results presented in Table 5 and 6, we observe that all temporal-oriented training schemes enhance the temporal understanding capabilities of LVLMs. Notably, the 13B-LVLM shows a more pronounced improvement, particularly when trained with the aggregated scheme VC+MC+MG+DC. This indicates significant untapped potential for even larger LVLMs, especially those exceeding the 20B parameter scale. In this work, we aggregate datasets across VC, MC, MG, and DC tasks, ensuring that each task type is presented to the model in a balanced manner. Due to computational constraint, we leave further exploration of the potential importance of task ordering in future work.

*Takeaway 2: For the remaining experiments, we add an additional temporal-oriented training stage and use VC, MC, MG, and DC as training schemes.*

Table 5: Effects of Temporal-Oriented Training Schemes on 7B-LVLM

| Training Scheme | MSRVTT | MSVD | ActivityNet-QA | Breakfast | COIN | LVU |
|---|---|---|---|---|---|---|
| No temporal training | 49.4 | 60.8 | 50.6 | 93.1 | 93.4 | 63.8 |
| VC | 50.3 | 61.5 | 51.0 | 93.2 | 93.7 | 64.4 |
| MC | 51.9 | 62.9 | 52.0 | 93.9 | 93.9 | 64.5 |
| MG | 50.9 | 62.3 | 51.4 | 93.3 | 93.9 | 64.6 |
| DC | 53.1 | 64.3 | 51.2 | 93.5 | 93.6 | 64.1 |
| VC + MC + MG + DC | **54.5** | **66.4** | **52.4** | **93.7** | **94.1** | **65.5** |

Table 6: Effects of Temporal-Oriented Training Schemes on 13B-LVLM

| Training Scheme | MSRVTT | MSVD | ActivityNet-QA | Breakfast | COIN | LVU |
|---|---|---|---|---|---|---|
| No temporal training | 60.6 | 74.3 | 52.5 | 93.7 | 93.2 | 71.8 |
| VC | 62.0 | 74.8 | 53.9 | 94.0 | 93.6 | 72.5 |
| MC | 61.2 | 74.6 | 53.0 | 93.5 | 93.9 | 71.9 |
| MG | 61.8 | 74.6 | 53.4 | 93.6 | 93.5 | 72.1 |
| DC | 62.2 | 75.3 | 54.3 | 94.7 | 94.0 | 72.7 |
| VC + MC + MG + DC | **62.7** | **75.8** | **54.5** | **95.1** | **94.7** | **72.8** |

### STEP 3: MEMORY BANK FOR VIDEO REPRESENTATIONS

Building upon the model developed in Step 2, we investigate how an LVLM processes video inputs. A straightforward approach is encoding visual frames or patches and concatenating their representations along the temporal axis. However, limited context length limit of the LVLM, coupled with GPU memory constraints, restricts the number of video frames that can be processed simultaneously. An alternative strategy is to apply temporal pooling (Maaz et al., 2023; Xu et al., 2024), but as demonstrated in our Step 1 analysis, this leads to suboptimal performance. Instead, we propose a different approach, *i.e.* processing video frames sequentially and storing their features in a memory bank. We conduct an ablation study on the size of the memory bank $B \in \{10, 20, 30, 40\}$ and present our findings in Table 8 and 9.

Based on these results, we find that incorporating a memory bank is an effective strategy, consistently outperforming standard pooling methods. Randomly sampling a fixed number of frames also proves suboptimal, particularly for long-term temporal understanding, as the sampled frames might fail to capture critical video context. Lastly, we note that increasing the memory bank size yields more significant improvement for the 13B-LVLM than the 7B-LVLM. This indicates that larger-scale models possess greater capacity to absorb and utilize richer video information.

*Takeaway 3: For our remaining experiments, we add a memory bank for video encoding.*

### STEP 4: MIXTURE-OF-EXPERTS FOR Q-FORMER

Building upon Step 3, we next explore strategies to enhance the capacity of the vision-language interface, which plays a critical role in conveying video information to the LLM. Given that naively adding randomly initialized layers tends to yield suboptimal performance, as demonstrated in Step 1, we turn to the mixture-of-experts (MoE) approach. An MoE module consists of a router and a set of experts, where each expert is a feedforward network. The router typically comprises a linear projection followed by a gating function, *e.g.* ReLU or Softmax, to compute the probabilities for routing a query token to specific experts. When a token encounters the MoE, the router selects a subset of experts to process the token, and their outputs are combined additively. This technique allows us to expand the parameter capacity of the Q-Former while keeping computational cost and latency manageable, as the model activates only a fraction of the total parameters for each token.

The exploration of MoE has remained scarce for LVLMs, especially for the vision-language interface, even though it has been investigated extensively in LLMs (Cai et al., 2024). In our work, we will experiment with the following categories of MoE:

- **Dense MoE:** the dense MoE activates all expert networks during each iteration. Based on the probability that the router produces for each expert, the outputs for an input token will be aggregated accordingly.
- **Sparse MoE:** to reduce computational overhead, we can activate only a subset of experts during each forward pass. To achieve this sparsity, we can compute a weighted sum of

Table 7: Examples of temporal-oriented training schemes.

| Video | Training Scheme | Example prompt | Sample Expected Output |
|---|---|---|---|
|  | VC | What does the video describe? | the buffalo bills new stadium is discussed as being deemed ineffective and not worth the investments made by the city and state. |
|  | MC | Explain what happened from frame 1 to frame 4 in the video. | woman in long white dress walking up a hillside path. |
| | MG | During which frames in the video can we observe "*woman in long white dress walking up a hillside path*"? | from frame 1 to frame 4 |
| | DC | Can you give me a breakdown of the occurrences at different timestamps in the video? | woman in long white dress walking up a hillside path, from 1 to 4. a woman sitting on the beach with long hair, from 5 to 8. [...] |

Table 8: Effect of Memory Bank on 7B-LVLM

| Memory Bank ($B$) | MSRVTT | MSVD | ActivityNet-QA | Breakfast | COIN | LVU |
|---|---|---|---|---|---|---|
| $B = 0$ | 54.5 | 66.4 | 51.4 | 93.7 | 94.1 | 65.5 |
| $B = 10$ | 56.2 | 68.2 | 51.9 | 93.9 | 94.1 | 66.0 |
| $B = 20$ | 60.7 | 72.5 | 52.5 | 94.3 | 94.7 | 68.1 |
| $B = 30$ | 60.7 | 72.5 | 52.5 | 94.7 | 94.6 | 68.1 |
| $B = 40$ | 60.7 | 72.6 | 52.5 | 94.9 | 94.8 | 68.2 |

the expert outputs from only the top-$k$ experts, rather than combining the outputs from all experts. In our work, we experiment with top-$k$ where $k = 1$ or $k = 2$.

For each type of MoE, we ablate the number of experts $E \in \{2, 4, 8\}$. In addition to Q-Former, we also add MoE to LLM to comprehensively study its effect on the LVLM.

Based on the results in Table 10 and 11, we observe that integrating MoE into Q-Former leads to a substantial boost in video understanding performance. Moreover, we note that both sparse and dense MoE categories bring improvement, with sparse MoE being slightly more effective. We hypothesize that sparse MoE provides a higher degree of specialization for LVLM to handle specific types of temporal circumstances. Perhaps unsurprisingly, scaling up MoE with more experts puts more significant impact to the 13B-LVLM than the 7B-LVLM, which implies further potential for LVLM in the upscaling direction.

On the other hand, adding MoE to LLM degrades the performance. We hypothesize that (1) adding MoEs can exacerbate attention decay over long contexts (Yao et al., 2024), reducing the model's ability to integrate temporally distant events, and (2) MoEs might tamper with the pre-trained knowledge in LLM.

*Final takeaway: Our final scaled-up temporal-oriented LVLM improves the initial LVLM baseline by 10.1% and 5.1% in terms of the 7B and 13B variant, respectively.*

## 4 EXPERIMENTAL RESULTS

We validate our temporal-oriented recipe on three popular video understanding tasks, *i.e.* video question answering, video captioning, and temporal grounding. For implementation details, baseline descriptions, and dataset descriptions, we refer our readers to Appendix.

**Video question answering.** We compare our results with existing methods on on recent datasets, *i.e.* MVBench (Li et al., 2024a), TempCompass (Liu et al., 2024b), VideoMME (Fu et al., 2025), and MLVU (Zhou et al., 2024). They are all designed to more comprehensively challenge video understanding capacity of multimodal models. Zero-shot results in Table 12 demonstrate our advantage over previous models, validating generalizability of our models' temporal understanding capacity.

**Temporal grounding.** To validate whether the model trained using our recipe genuinely possesses temporal understanding capacity, we decide to experiment with the temporal grounding task on the Charades-STA dataset (Gao et al., 2017). As shown in Table 13, our model achieves on-par performance with models specialized in temporal grounding (Zhang et al., 2019; 2020; Zeng et al., 2020; Lei et al., 2021), and outperforms generic video LLMs (Li et al., 2024a; Maaz et al., 2023; Zhang et al., 2023). This showcases our training recipe is able to touch the fundamental component that impacts temporal learning of LVLMs.

Table 9: Effect of Memory Bank on 13B-LVLM

| Memory Bank ($B$) | MSRVTT | MSVD | ActivityNet-QA | Breakfast | COIN | LVU |
|---|---|---|---|---|---|---|
| $B = 0$ | 62.7 | 75.8 | 54.5 | 95.1 | 94.7 | 72.8 |
| $B = 10$ | 63.3 | 75.9 | 54.9 | 95.3 | 94.9 | 73.1 |
| $B = 20$ | 63.5 | 76.3 | 55.2 | 96.0 | 95.0 | 73.4 |
| $B = 30$ | 63.5 | 76.3 | 55.4 | 96.5 | 95.6 | 73.9 |
| $B = 40$ | **64.4** | **77.5** | **55.8** | **97.6** | **96.8** | **74.7** |

Table 10: Effect of Mixture-of-Experts (MoE) on 7B-LVLM

| Position of MoE | Type | Experts ($E$) | MSRVTT | MSVD | ActivityNet-QA | Breakfast | COIN | LVU |
|---|---|---|---|---|---|---|---|---|
| Q-Former | Sparse | 2 | 63.9 | 76.9 | 56.3 | 95.6 | 95.4 | 72.9 |
|  |  | 4 | 64.1 | 77.6 | 56.7 | 96.3 | 96.1 | 73.7 |
|  |  | 8 | **65.0** | **78.1** | **57.3** | **97.0** | **96.6** | **74.5** |
|  | Dense | 2 | 63.9 | 76.4 | 55.7 | 95.2 | 94.9 | 72.3 |
|  |  | 4 | 64.4 | 76.8 | 56.6 | 96.0 | 95.8 | 73.0 |
|  |  | 8 | 64.9 | 77.3 | 57.0 | 96.1 | 96.2 | 73.4 |
| LLM | Sparse | 2 | 54.9 | 72.9 | 55.4 | 93.4 | 92.0 | 72.5 |
|  |  | 4 | 54.9 | 73.5 | 55.5 | 93.6 | 92.2 | 72.7 |
|  |  | 8 | 55.4 | 73.6 | 55.9 | 93.6 | 92.5 | 72.8 |
|  | Dense | 2 | 54.5 | 72.4 | 54.9 | 93.1 | 91.7 | 72.4 |
|  |  | 4 | 54.8 | 72.7 | 55.2 | 93.4 | 91.8 | 72.6 |
|  |  | 8 | 54.9 | 73.1 | 55.6 | 93.7 | 92.0 | 72.9 |

Table 11: Effect of Mixture-of-Experts (MoE) on 13B-LVLM

| Position of MoE | Type | Experts ($E$) | MSRVTT | MSVD | ActivityNet-QA | Breakfast | COIN | LVU |
|---|---|---|---|---|---|---|---|---|
| Q-Former | Sparse | 2 | 65.2 | 78.4 | 57.2 | 97.6 | 97.5 | 75.5 |
|  |  | 4 | 65.6 | 78.8 | 57.6 | 97.8 | 97.7 | 75.9 |
|  |  | 8 | **66.7** | **79.5** | **58.3** | **98.5** | **97.8** | **76.1** |
|  | Dense | 2 | 65.0 | 77.3 | 56.1 | 96.4 | 96.5 | 74.7 |
|  |  | 4 | 65.5 | 77.7 | 56.8 | 96.6 | 96.8 | 74.9 |
|  |  | 8 | 66.2 | 78.3 | 57.3 | 97.6 | 97.0 | 75.7 |
| LLM | Sparse | 2 | 59.3 | 73.3 | 53.4 | 91.9 | 92.9 | 69.1 |
|  |  | 4 | 59.8 | 73.6 | 53.7 | 93.1 | 93.6 | 69.2 |
|  |  | 8 | 60.2 | 73.7 | 54.1 | 93.2 | 94.2 | 69.6 |
|  | Dense | 2 | 58.7 | 72.7 | 52.9 | 92.2 | 92.3 | 71.9 |
|  |  | 4 | 59.0 | 72.9 | 53.0 | 92.3 | 92.5 | 70.0 |
|  |  | 8 | 59.4 | 73.7 | 53.9 | 92.3 | 93.1 | 70.4 |

Table 12: Comparison on MVBench, TempCompass, VideoMME, and MLVU.

| Method | MVBench | TempComp. | VideoMME | MLVU |
|---|---|---|---|---|
| LLaMA-VID (Li et al., 2024b) | – | 34.8 | – | – |
| VideoChat2 (Li et al., 2024a) | 51.1 | 38.5 | – | 29.2 |
| Video-ChatGPT (Maaz et al., 2023) | 32.7 | 31.8 | – | 31.3 |
| 7B-PLLaVA (Xu et al., 2024) | 50.1 | – | – | – |
| 13B-PLLaVA (Xu et al., 2024) | 58.1 | – | – | – |
| ST-LLM (Liu et al., 2024a) | 54.9 | – | 42.3 | – |
| VALLEY (Luo et al., 2023) | – | 26.3 | – | – |
| MA-LMM (He et al., 2024) | – | – | – | 36.4 |
| 7B-LVLM (Ours) | 60.2 | 43.7 | 45.5 | 42.1 |
| **13B-LVLM (Ours)** | **61.3** | **44.4** | **46.3** | **42.8** |

Table 13: Comparison on Charades-STA.

| Method | R1@0.5 | R1@0.7 | R5@0.5 | R5@0.7 |
|---|---|---|---|---|
| MAN (Zhang et al., 2019) | 41.2 | 20.5 | 83.2 | 51.9 |
| 2D-TAN (Zhang et al., 2020) | 39.7 | 23.2 | 80.3 | 51.3 |
| DRN (Zeng et al., 2020) | 42.9 | 23.7 | 87.8 | 54.9 |
| RaNet (Gao et al., 2021) | 43.9 | 26.8 | 86.7 | 54.2 |
| Moment-DETR (Lei et al., 2021) | 49.0 | 21.2 | 87.0 | 50.1 |
| UMT (Liu et al., 2022) | 49.4 | 26.2 | 89.4 | 55.0 |
| VideoChat2 (Li et al., 2024a) | 3.3 | 1.3 | – | – |
| Video-LLaMA (Zhang et al., 2023) | 3.8 | 0.9 | – | – |
| Video-ChatGPT (Maaz et al., 2023) | 7.7 | 1.7 | – | – |
| **7B-LVLM (Ours)** | 46.6 | 23.8 | 86.8 | 52.0 |
| **13B-LVLM (Ours)** | 47.4 | 24.7 | 87.2 | 52.5 |

## 5 CONCLUSION

In this work, we highlight the critical role of temporal modeling in the design of modern LVLMs. In particular, we discover that key components, including query transformer (Q-Former), temporal-oriented training schemes, memory bank, and MoE augmentation for Q-Former, are pivotal for effective video understanding with LVLMs. Our empirical findings culminate in a step-by-step, temporal-oriented recipe for constructing effective temporal modeling capacity in LVLM. Compared with existing LVLMs, our proposed approach achieves superior performance across a broad range of standard video understanding datasets. Notably, the benefits of our recipe become more pronounced for larger-scale LVLMs, underscoring the potential of explicitly incorporating temporal modeling into large-scale architectures.

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

# A    IMPLEMENTATION DETAILS

## VIDEO QUESTION ANSWERING

We formulate VideoQA as a token generation task. After conducting the temporal-oriented training stage, we fine-tune the model to optimize its performance on each downstream dataset, using the averaged cross-entropy loss of each token between the generated answer and the groundtruth answer.

## VIDEO CAPTIONING

Since the nature of the task is inherently token generation, the only remaining concern is the evaluation protocol. Because strictly adhering to surface words might not adequately assess model quality, we follow existing LVLM works (Lin et al., 2023; Maaz et al., 2023) to use *gpt-3.5-turbo* to judge the quality of the answer.

## A.1    TEMPORAL GROUNDING

We also formulate temporal grounding as a token generation task. Specifically, our models generate the answer to denote the start and end frame index of the grounded moment, then we map the indices into real timestamps. Following temporal grounding literature (Nguyen et al., 2023b), we use Recall@1{0.5,0.7} and Recall@5{0.5, 0.7} as evaluation metrics.

# B  BASELINE DESCRIPTIONS

- **Video-UTR** (Yu et al., 2025): a video MLLM instruction-finetuned with a temporal hacking reward
- **LLaVA-NeXT** (Zhang et al., 2024): a MLLM that uses a grid-based encoding strategy and is trained on high-quality multimodal data mixture.
- **MA-LMM** (He et al., 2024): a MLLM which encodes video frames in an online manner to circumvent the LLMs' context length limits.
- **VALLEY** (Luo et al., 2023): a MLLM whose training procedure is equipped with diverse and enhanced video-text alignment data.
- **LLaMA-VID** (Li et al., 2024b): uses two distinct tokens, including context and content token, to resolve large computational burdens of long videos.
- **VideoChat2** (Li et al., 2024a): a video MLLM that is trained with a progressive multimodal training procedure with diverse instruction-tuning data.
- **Video-ChatGPT** (Maaz et al., 2023): a multimodal model that trains a linear layer to map visual encoded representations into the LLM's language space.
- **Video-LLaVA** (Lin et al., 2023): exhibits a similar paradigm to Video-ChatGPT but initializes visual encoder with LanguageBind (Zhu et al., 2023) and unfreezes its linear vision-language interface with the LLM during training.
- **GPT4Video** (Wang et al., 2024b): a MLLM that is instruction-following fine-tuned along with a stable diffusion generative model.
- **PLLaVA** (Xu et al., 2024): a LLaVA model extension to video understanding with an appropriate adaptive pooling layer integrated into the vision-language interface.
- **ST-LLM** (Liu et al., 2024a): a MLLM which is fed all spatial-temporal tokens into its LLM and uses a dynamic masking strategy to overcome the overhead and instability issues of such full-token sequence.
- **Chat-UniVi** (Jin et al., 2024): employs a set of dynamic visual tokens to uniformly represent images and videos.
- **Video-LLaMA** (Zhang et al., 2023): integrates audio signals into the MLLM and uses an Audio Q-Former to assemble a pre-trained image encoder into video encoder.
- **MAN** (Zhang et al., 2019): utilizes a structured graph network to model video moment-wise temporal relationships.
- **2D-TAN** (Zhang et al., 2020): projects video moments onto a 2D map where column and row indices represent starting and ending points, respectively.
- **DRN** (Zeng et al., 2020): a dense regression network which considers all video moments as positive samples and seeks to predict their distances to the ground-truth starting and ending boundaries.
- **RaNet** (Gao et al., 2021): comprises a BiLSTM language encoder, a CNN image encoder, and a cross-modal graph to capture moment-query relations and select target moments.
- **Moment-DETR** (Lei et al., 2021): a specialized temporal grounding model with a transformer-based encoder-decoder model that views the problem as a direct set prediction problem.
- **UMT** (Liu et al., 2022): a multimodal transformer model that is targeted at temporal grounding using either audio or text queries.

# C  DATASET DESCRIPTIONS

TEMPORAL-ORIENTED TRAINING

We conduct an additional temporal-oriented training stage after the model has been pretrained and instruction-tuned in previous works. The datasets we use consist of InternVid (Wang et al., 2023) and VIDAL-10M (Zhu et al., 2023).

- **InternVid (745K):** the original dataset comprises 234M video clips accompanied by detailed descriptions from 7M videos. Due to computational and storage limit, we use only 745K clips to train our model.

- **VIDAL-10M (661K):** consists of 10M short videos paired with corresponding descriptions. In our work, we utilize 661K videos to train our LVLM.

### VIDEO QUESTION ANSWERING

We evaluate on three short-term VideoQA datasets, *i.e.* MSRVTT (Xu et al., 2016), MSVD (Chen & Dolan, 2011), and ActivityNet-QA (Caba Heilbron et al., 2015), and three long-term VideoQA datasets, *i.e.* Breakfast (Kuehne et al., 2014), COIN (Tang et al., 2019), and LVU (Wu & Krahenbuhl, 2021).

- **MSRVTT** (Xu et al., 2016) composed of 10K YouTube videos, for VideoQA the dataset is formatted into 243K open-ended questions. We adopt the 149K-12K-73K train-val-test split to evaluate LVLMs.

- **MSVD** (Chen & Dolan, 2011) comprises 47K open-ended questions for 2K videos. We employ a split of 30K/6K/13K to divide the questions into training, validation, and testing sets, respectively.

- **ActivityNet-QA** (Caba Heilbron et al., 2015) consists of 58K open-ended questions on 5.8K sampled videos from ActivityNet (Caba Heilbron et al., 2015).

- **Breakfast** (Kuehne et al., 2014) encompasses 1.7K videos related to 10 actions for breakfast preparation. The model is asked to predict the action type in the video.

- **COIN** (Tang et al., 2019) includes 12K videos from YouTube, covering 180 diverse tasks in 12 domains related to daily life. The model is tasked with predicting the task type conducted in the video.

- **LVU** (Wu & Krahenbuhl, 2021) consists of 30K videos sourced from 3K movies. Given a video, we train/test the model to predict the relationship, speaking style, scene, director, genre, writer, and release year of the video.

- **MVBench** (Li et al., 2024a) consists of 3.7K QA pairs across 11 public video benchmarks. These testing pairs cover scenarios that the model cannot resolve by relying on a single frame.

- **TempCompass** (Liu et al., 2024b) comprises videos with 7.5k diverse temporal compositional questions designed to test a model's ability to understand complex temporal aspects, *i.e.* action, speed, direction, attribute change, and event order.

- **VideoMME** (Fu et al., 2025): possesses 900 diverse, manually annotated videos, totally 256 hours of content, accompanied by 2700 QA pairs. We conduct zero-shot evaluation of our models trained using the proposed recipe on such 2700 QA pairs.

- **MLVU** (Zhou et al., 2024): comprises 3.1K QA pairs related to 1.7K videos of various genres and duration levels. These QA pairs cover close-ended and open-ended tasks, span multiple dimensions of LVU tasks, and involve local information with clear referring context.

### VIDEO CAPTIONING

We evaluate our model on two prevalently used datasets, *i.e.* MSRVTT (Xu et al., 2016) and MSVD (Chen & Dolan, 2011).

- **MSRVTT** (Xu et al., 2016) consists of 200K videos paired with respective captions. To ensure fair comparison with previous works (Li et al., 2024b; Nguyen et al., 2024a; He et al., 2024), we use a split ratio of 130K/10K/60K for training, validation, and testing.

- **MSVD** (Chen & Dolan, 2011) contains 81K videos and corresponding captions. Following recent works (Li et al., 2024b; Nguyen et al., 2024a; He et al., 2024), we adopt a ratio of 49K/4K/28K to split these samples into training, validation, and testing sets.

TEMPORAL GROUNDING

We evaluate our model on a well-known popular dataset for temporal grounding, *i.e.* Charades-STA (Gao et al., 2017).

- **Charades-STA** (Gao et al., 2017): consists of videos about daily indoor activities. The dataset is split into 12,408 and 3,720 moment annotations for training and testing. We only use the testing subset to evaluate our models.

