# OpenReview forum: "Temporal-Oriented Recipe for Transferring Large Vision-Language Model to Video Understanding"
_ICLR.cc/2026/Conference — Submitted to ICLR 2026_

### Official Review · Reviewer_hUuN · 2025-10-29

**Soundness:** 3
**Presentation:** 3
**Contribution:** 2
**Rating:** 4
**Confidence:** 5

**Summary:**

Summary:

- Key Idea: This paper addressed the challenge of enabling image-pretrained LVLMs to acquire temporal understanding for video reasoning. This paper proposes a systematic empirical framework called the Temporal-Oriented Recipe, which consists of 4 modular steps:

-- Step 1: Vision-Language Interface Optimization

-- Step 2: Temporal-Oriented Intermediate Training

-- Step 3: Memory Bank

-- Step 4: Mixture-of-Experts

- Result: Compared to the baseline LVLM approach discussed in this paper, the complete 4-step pipeline achieves single-digit to double-digit accuracy improvements on standard VideoQA benchmarks. Typical cumulative gains range from approximately 6% to 12% percentage points, with specific values depending on the dataset and backbone network size. Zero-shot evaluations on MVBench/TempCompass/VideoMME/MLVU demonstrate that this approach achieves gains over baseline LVLM in categories emphasizing temporal order and causal relationships.

**Strengths:**

- The paper directly addresses LVLMs’ weakness in temporal reasoning and offers a modular, interpretable solution that practitioners can readily adopt.

- Comprehensive Component Comparison: The authors clearly analyze trade-offs between interface types, temporal training, memory, and MoE design, revealing practical engineering guidelines (for instance, “Q-former > Linear”, “MoE benefits interfaces but not LLM”).

- Well-structured temporal tasks: The combination of VC/MC/MG/DC tasks covers a broad range of temporal semantics, and their effects have been empirically demonstrated.

- Extensive Zero-shot Evaluation: The model demonstrates strong performance across 4 major video benchmarks, achieving good results on both QA and localization tasks.

**Weaknesses:**

- Lack of Novelty: This paper combines existing modules: Q-former, temporal multi-tasking training, memory banks, and MoE, but introduces no new architecture or learning objective. While the “recipe” framing is practical, it’s an engineering integration (not really novel) rather than a conceptual breakthrough.

-	Limited theoretical explanation for observed results: This paper reports empirical gains (for instance, “MoE helps at the interface but not in LLM), but gives no real analytical justification, which makes the findings more observational than scientifically grounded.

-	Weak justification of design decisions: The selection of Linear vs Q-former, or Q-former vs BERT-init Q-former, is tested, but the reason for choosing only these options is not clearly motivated. Other interface mechanisms are not compared (for example, cross-attention fusion, perceiver-style adapter).

-	Questionable claim of revising prior findings: The paper asserts that it revises earlier conclusions about the sufficiency of linear projection. However, since the authors did not reproduce or align with the original experimental setups, this statement remains suggestive rather than conclusive.

-	Lack of Ablation:

1.	No pipeline order ablation: The paper follows a fixed sequence (Interface -> Temporal Training -> Memory -> MoE) without verifying whether this order is optimal or random. Alternative orders (like inserting memory earlier) are not explored.

2.	Insufficient multi-task ablation (VC+MC+MG+DC): Only single-task and four-task joint training results are reported; partial combination and task weighting sensitivity are missing, leaving unclear which task contributes most.

3.	Limited memory bank scaling analysis: Experiments terminated at B=40 without exploring adaptive memory sizes relative to video length or computational trade-offs.

4.	Limited MoE hyperparameter sweeps: Key routing parameters (for example, top-k) are also missing.

-	Unclear quantitative contribution of each component: Although the paper is structured step-by-step, there is no single summary table showing cumulative performance from step 0 to step 4. Readers cannot easily see which step contributes most or least.

-	Incomplete SOTA coverage and missing entries: Several tables contains blank results on some datasets. For example, in Table 12, only one SOTA model is compared against the proposed model on the VideoMME dataset, which is insufficient.

-	Lack of comparison with specialized temporal reasoning models: The paper primarily compares with general LVLMs, rather than temporal-specific models, which limits the strength of its claim that the proposed “recipe” improves temporal reasoning rather than general video comprehension. Also, it doesn’t clarify whether it complements or replaces existing temporal reasoning models. No discussion of conceptual connections and differences from these SOTA temporal modeling frameworks. (for example, ST-LLM, MovieChat…).

-	Short-video bias weakens temporal claims: Most tested datasets contain short clips. Models focused on long-range temporal coherence are not compared (for example, Epic-Kitchen, Ego4D, YouCook2, LongVideoBench…).

-	Limited temporal metrics: Evaluation mainly reports overall accuracy or GPT-based caption scores, without temporal consistency.

-	Computational and latency metrics are not reported: The added Q-former layer, memory bank, and MoE significantly increase parameter and inference costs, but GPU runtime, memory usage, or latency are not documented.

**Questions:**

Could you clarify your main contributions?

---

### Official Review · Reviewer_fuoT · 2025-10-31

**Soundness:** 2
**Presentation:** 2
**Contribution:** 2
**Rating:** 2
**Confidence:** 5

**Summary:**

This paper proposes a training recipe for adapting image-trained LVLMs to temporally dependent video understanding tasks. The author’s key motivation is that existing LVLMs do not explicitly model temporal relationships, limiting their ability to perform well on temporally dependent tasks. To address this, they explore both the architecture (Q-Former, memory bank, MOE Q-Former) and data choices (temporally oriented instruction pairs) when fine-tuning LVLMs. Each component of their training scheme is evaluated independently to optimize its design, the resulting temporally oriented training recipe is a combination of these components.

**Strengths:**

The work is well motivated as existing LVLMs often struggle on temporally dependent video understanding tasks. The ability to improve temporal modeling through a unified training recipe would be desirable to the community

The authors provide a comprehensive evaluation of the LVLM components explored in the work. They also explore components like memory banks, which are relatively under explored in LVLMs

The proposed training recipe outperforms existing pre-trained LVLMs

**Weaknesses:**

It isn’t well explored why the proposed recipe improves temporal modeling capabilities of LVLMs. A deeper analysis outside of purely empirical results would strengthen the argument (e.g., attention visualization)
* Specifically the Q-Former and MOE augmented Q-Former, it is unintuitive to the reviewer why these components in particular improve temporal understanding. It could be the case that these components simply enhance general video understanding rather than temporal understanding

The novelty of the proposed recipe is limited. Similar approaches such as Apollo (Zohar et al., Apollo: An Exploration of Video Understanding in Large Multimodal Models, CVPR 2025) also propose training recipes for improving temporal understanding in LVLMs and investigate many of the same components (vision-language connector, fine-tuning data)

The backbone LVLM components used and evaluated in this work (e.g., CLIP and Vicuna) are dated and rarely used these days (in contrast to SigLIP and Qwen), limiting the applicability of the proposed training recipe

From Line 204, it is unclear if the base image-trained VLM is BLIP 2 or LLaVA

Concerns on performance improvements in Table 12 and Table 13
* Table 12: While the proposed 7B and 13B LVLMs improve results over existing LVLMs, the results are sparse (~60% of the table is empty) making it difficult to gauge whether the proposed method consistently improves performance
* Table 13: The LVLM models seem to be evaluated zero-shot which is unfair as they were not trained on temporal grounding data. It makes more sense to evaluate LVLMs like TimeChat (Ren et al., TimeChat: A Time-sensitive Multimodal Large Language Model for Long Video Understanding, CVPR 2024) which are trained with temporal grounding data. The other methods are not LVLMs and do not help convince the reviewer that the proposed LVLM training recipe is preferable to basic LVLM training recipes
* There is no comparable baseline in either of the tables. It seems like the base model used by the authors is BLIP 2 (see weakness above regarding Line 204), in this case a fair baseline would be fine-tuning of the base BLIP 2 model on the corresponding video data
* Some of the baselines in Appendix B don’t appear to be used as baselines in the paper, for example LLaVA-NeXT and Video-UTR

Minor formatting comments:
* In Table 2 “Training Data Size (IFT)”, IFT is never defined in the text, but I assume it stands for instruction-following fine-tuning
* The table headers in Table 12 and Table 13 should be bolded
* In Table 12 the best and 2nd best results are bolded, but in Table 13 no results are bolded

**Questions:**

Can the authors provide evidence showing that their proposed recipe truly enhances temporal modeling (e.g., through attention maps or feature visualizations)?

Why do the Q-Former and MOE-augmented Q-Former improve temporal understanding rather than general video comprehension?

How does this training recipe differ from prior works like Apollo that also propose training recipes for temporal understanding in LVLMs?

Is the base vision-language model used for fine-tuning BLIP-2 or LLaVA? If it is BLIP 2, how is the Q-Former swapped with a linear layer for the “Linear projection” results in Table 3 and Table 4?

Can the authors provide fair and complete baseline comparisons, including equivalent fine-tuned base models and LVLMs trained with temporal grounding data?

The source of the Breakfast and COIN benchmarks (for VLMs) is unclear. The reviewer is aware of these datasets, but not of any existing approaches that convert them to an instruction format to evaluate them with VLMs. The VLMs used for comparison also do not seem to evaluate on Breakfast or COIN. Could the authors clarify where these benchmarks come from?

---

### Official Review · Reviewer_oj2v · 2025-11-01

**Soundness:** 3
**Presentation:** 3
**Contribution:** 2
**Rating:** 4
**Confidence:** 4

**Summary:**

This paper presents a systematic, empirically-driven "recipe" to enhance temporal understanding in Large Vision-Language Models (LVLMs) by sequentially evaluating and integrating existing components—a pre-trained Q-Former as the vision-language interface, an additional training stage with temporal tasks (VC, MC, MG, DC), a video memory bank, and a Mixture-of-Experts augmented Q-Former. While the proposed combination demonstrates consistent performance gains across multiple video QA and temporal grounding benchmarks, the work primarily serves as a validation of a high-performing configuration rather than introducing novel mechanisms, lacking deeper analysis into why these specific components interact synergistically and failing to establish strong comparisons against the most recent state-of-the-art models.

**Strengths:**

+ This paper is easy to follow.

+ The step-by-step empirical investigation is commendable. The ablation studies for each component are clear and provide a strong, reproducible blueprint for building temporally-aware LVLMs.

+ The paper validates its recipe across multiple tasks (VideoQA, Captioning, Temporal Grounding) and datasets, including both short-term and long-term video understanding benchmarks, demonstrating the generalizability of the approach.

**Weaknesses:**

1. Novelty:

- The core contribution is the combination of existing components rather than the invention of new ones. Q-Former, Memory Banks, MoE, and the temporal training objectives (VC, MC, MG, DC) are all established techniques. The contribution is primarily the empirical finding that this specific combination works well.

2. Theoretical Analysis:

- There is no deeper analysis of how the Q-Former, memory bank, and MoE interact. Does the memory bank alleviate a specific bottleneck in the Q-Former? Do the experts in the MoE specialize in different temporal phenomena (e.g., fast vs. slow actions, order vs. duration)?

- What specific temporal skills are improved? The evaluation is based on end-task performance. A diagnostic analysis is missing: does the model get better at reasoning about order, duration, causality, or all of them? Visualizations of attention maps or analysis of MoE routing could have provided crucial insights.

- The "recipe" lacks a unifying principle. It is a collection of parts without a theoretical foundation explaining their synergy in capturing temporal dynamics.

3. Writing:

- The paper is padded with an excessive number of highly similar tables (Tables 3, 4, 5, 6, 8, 9, 10, 11) that show incremental results for 7B and 13B models. This occupies a substantial portion of the content while conveying a simple, repetitive message: "more layers/data/experts are better, and 13B is better than 7B." This space could have been better used for deeper analysis by consolidating results into summary graphs and moving detailed tables to an appendix.

**Questions:**

The full recipe appears computationally intensive. What is the inference time of the final model compared to the baselines?

---

### Official Review · Reviewer_b39P · 2025-11-11

**Soundness:** 2
**Presentation:** 1
**Contribution:** 1
**Rating:** 2
**Confidence:** 4

**Summary:**

The paper presents a systematic, step-by-step methodology, termed a "temporal-oriented recipe," to enhance the temporal understanding capabilities of Multimodal LLMs (MLLMs). The authors argue that most existing MLMMs implicitly handle time by relying on spatial reasoning, which limits their performance on tasks requiring an understanding of order, causality, and temporal relationships. They outline a standardized recipe, which involves a Vision-Language Interface using a Q-Former, with various settings, Temporal-Oriented Training on video datasets, a Memory Bank, and an MoE layer. The authors conduct extensive experiments on 7B and 13B parameter models, demonstrating that each step of their recipe yields progressive improvements. Their final model achieves improvements over the baseline across a range of video understanding benchmarks, including video question answering, temporal grounding, and captioning.

**Strengths:**

**Addresses a Critical Problem**: The work tackles a fundamental and widely recognized weakness of temporal reasoning in video-language models. Improving performance on causality, action sequencing, and long-range dependencies is a crucial step for the field, and in general discovering better techniques for modeling. This makes the topic at hand very important.

**Focus on Ablations**: The authors are very focused on performing detailed ablations, which is appreciated. Each step is compared against previous steps so that contributions can be clearly seen to be better than previous improvements.

**Weaknesses:**

**Limited Novelty of Recipe Design and Ablations**: As is, the work reads more as an extensive ablation study than a conference ready publication, and that a few different ideas were tried and then stitched together with no cohesion. The paper's contribution lies in the novel combination and empirical validation of existing techniques rather than the introduction of a new method. Q-Formers, memory banks, MoE, and temporal captioning/grounding tasks are all established concepts. As such, the work could be perceived as a large-scale engineering effort or an extensive ablation study rather than a paper presenting a fundamental new algorithm or architecture.

**Poor Comparisons Between Recipe Parts**: The comparisons done in the paper are not accurate apples to apples comparisons. In fact, the core basis for the beginning of the paper, Tables 3+4, only compare the effect of Q-Former to a Linear Projection. But of course a 12-layer Q-Former with 100M+ trainable parameters (in a standard case, not sure exactly here), compared to a linear projection with a few thousand parameters, will outperform it (and only barely to mention!). This needs to be a more fair comparison between the two adapters, as this forms the basis for the rest of the paper. In fact, most recent VLM papers find that Q-Former/Perceiver Resampler style token resampling is outperformed by simple spatial average pooling ([1], [2]). This part is not accurately accounted for either. Thus these ablations are incomplete. The data is again not matched, and is also added at a large scale (nearly 2M samples in just VC, MC and MG). For that scale, the gains are quite miniscule. In this case, “temporal training” is just some in distribution fine-tuning.

**Unorganized Information and Structure**: The paper is unorganized, which makes the key takeaway difficult to parse. The introduction did not clearly set up the paper for understanding the key points, the Tables 3+4 were difficult to make the appropriate comparisons in. Table 8 didn’t have takeaways written in the text, besides “more is better”, which actually seemed to plateau at B=20.

Overall, I consider these significant weaknesses which cannot be solved without revision. Therefore I strongly recommend this paper for rejection.

[1] Tong, Peter, et al. "Cambrian-1: A fully open, vision-centric exploration of multimodal llms." NeurIPS 2024.
[2] Chung, Jihoon, et al. "Unifying specialized visual encoders for video language models." ICML 2025.

**Questions:**

1. Actually, in Table 1, I would say the model’s responses being “late for exam,” “to the hospital,” and “feeling sad” may also point to strong language biases from pretraining being relied on more than visual cues (less so for “feeling sad”).
1. L201: Why is the sequence of frames randomly selected?
1. The introduction could be rewritten to focus concisely around the main point; it mentions a few different ideas like MLLM training differences, architecture differences, and empirical analysis, which I don’t entirely see a common theme through.

Most of my questions are in the weaknesses section already.

---

### Meta-Review · Area_Chair_yByv · 2026-01-05

**Summary:**

This paper studies temporal-oriented recipe to enhance the temporal understanding of VLMs. Their proposed recipe includes four steps: 1. vision-language interface; 2. Temporal-oriented training; 3. Adding memory back; 4. Adding mixture of experts.

Reviewers acknowledge that the proposed method 1. is well motivated and try to solve a critical problem, 2. provide comprehensive experiments and evaluations. Major concerns from reviewers are: 1. This paper has limited novelty, all techniques studied in this paper are existing techniques; 2. Lack of in-depth analysis or explanation for observed results; 3. The observations are not well structured, the authors present their results piece by piece and lacks a unified principle. Besides, reviewer b39P also pointed out that the comparison between recipes are not fair. These major concerns are valid so I tend to reject this paper.

**Reviewer Concerns:**

As there is not rebuttal posted by the authors, the concerns from the reviewers are not addressed.

**Reviewer Scores:**

This paper get scores of 2, 4, 2, 4. As no rebuttal posed by the authors, I tend to believe the reviewers will remain their score. The paper then gets an average score of 3 which is not qualified for publication.

---

### Decision · Program_Chairs · 2026-01-26

Reject